# How Does Job Well-Being Optimize Audit Performance? The Moderating Effect of Passion

**DOI:** 10.3390/bs15010042

**Published:** 2025-01-03

**Authors:** Kuo-Chih Cheng, Yuan-Sheng Lin, Tung-Chin Yang, Tsung-Fu Chuang, Hsiu-Mei Lai, Lan-Hui Lin, Shao-Hsi Chung

**Affiliations:** 1Department of Accounting, National Changhua University of Education, Changhua City 500, Taiwan; juice@cc.ncue.edu.tw (K.-C.C.); linlh@cc.ncue.edu.tw (L.-H.L.); 2Department of Finance, National Changhua University of Education, Changhua City 500, Taiwan; 11228@dfps.tp.edu.tw (Y.-S.L.); d0867008@mail.ncue.edu.tw (T.-C.Y.); d1067010@mail.ncue.edu.tw (T.-F.C.); d1067005@mail.ncue.edu.tw (H.-M.L.); 3Department of Business Administration, Meiho University, Pingtung County 912, Taiwan

**Keywords:** job well-being, audit performance, passion, affective events theory, CPA firm

## Abstract

Most prior studies found that job well-being and job performance are in a linear relationship. Audit firms are a type of highly professional organization. Based on the affective events theory, this study argues that job well-being can accelerate the improvement of audit performance thus forming a curvilinear relationship. Additionally, auditing is a job that demands highly professional skills and responsibility. This study argues that an auditor’s passion for work can strengthen the relationship between job well-being and audit performance. The study employed a quantitative survey approach, collecting data from 178 auditors who are in a position of in-charge, deputy manager, and manager in the Big Four CPA firms in Taiwan. The empirical evidence confirmed that job well-being and audit performance are in a curvilinear relationship. In addition, the effect of job well-being on audit performance is greater in the presence of high passion and will diminish when the passion is low. According to the research results, the managerial implications for audit firms are provided.

## 1. Introduction

Since the Enron collapse, audit firms around the globe have been faced with unprecedented challenges. While the audit costs and risks have drastically increased, audit fees have not been raised relatively. Therefore, to reevaluate auditors’ job and how to improve their audit performance has become an important issue for audit firms in recent years ([30]). The current research trends suggest that the relationship between job well-being and audit performance is not merely linear but may exhibit a curvilinear form, reflecting the complex interplay of workplace dynamics and psychological factors. Recent studies, such as the work by ([44]), support this view, indicating that both excessively high and low levels of job satisfaction can adversely affect job performance under certain conditions. To further comprehend the characteristics of auditing work and its impact on job well-being and audit performance, this study explores the daily work experiences of auditors through the lens of affective events theory. The auditing profession demands high levels of accuracy and responsibility, often accompanied by pressing deadlines and prolonged work pressures. These job characteristics can trigger emotional events, such as client interactions and challenges encountered during auditing discoveries, which may temporarily or persistently affect the emotional states and performance of auditors. By examining these professional demands through affective events theory, we can delve deeper into how these requirements test auditors’ job well-being and subsequently influence their audit performance.

Regarding the factors that affect the audit performance, in addition to the professional competence of the auditor, the personal feeling from the organization is also an important one ([37]; [42]). Former professor in Harvard Business School, David H. Maister, pointed out that greater employee well-being can lead to a higher product quality, better customer relationships, and a large improvement in the firm’s financial performance ([29]). Many empirical studies have confirmed that employees’ job performance is significantly related to their job well-being ([19]; [35]; [36]). In Taiwan, there are numerous companies which have achieved a higher job performance through an employee well-being policy. For example, Chi Mei Optoelectronics (renamed as InnoLux after merger) has always been regarded as a firm of well-being. It achieved a double-digit growth of earnings within only four years after implementing an employee well-being policy. It has created a successful model of using a well-being policy to deliver outstanding firm performance ([8]).

Based on affective events theory, employees’ emotional reactions may affect their job performance in the workplace with specific job characteristics or requirements ([2]). They argued that the initial emotional reactions caused by events at work usually do not last long, and employees’ job performance will become unstable with fluctuations of emotions. Influenced by positive events that continue to emerge in the workplace, employees’ positive emotions will develop into stable affect, which can, in turn, bring unexpected job performance. Based on this argument, we believe that job well-being and job performance are not simply in a linear relationship as suggested in previous research ([22]; [23]). Since auditors’ job is characterized by high professionalism and high responsibility, this study posits that job well-being and audit performance are in a curvilinear relationship, in which the effect of job well-being slightly decreases first and drastically increases later on the spectrum of audit performance. It is one main purpose of this study.

In addition, the relationship between personal performance and passion has been of concern for the practice and the academic, and constantly related to investigation and reporting. A survey conducted by Business Weekly showed that passion ranks first among the common leader traits of 200 global leaders ([9]). Passion is a common leadership characteristic in a survey of 50 outstanding CEOs ([33]). Moreover, passion in leaders can create job opportunities and lead to higher financial performance ([38]). There is a positive relationship between one’s passion for one’s job and the performance among the employees of an insurance company ([21]). Although audit performance may be influenced by the job well-being and passion of auditors, in prior studies on audit performance, the two factors have never been discussed in the same framework. For this sake, as another purpose of this study, we consider passion as a contextual variable in the relationship between job well-being and audit performance, to explore auditors with high passion or low passion and whether the relationship will be different. The results confirm the presence of a curvilinear relationship between job well-being and audit performance, challenging the traditional linear assumptions. Additionally, the moderating role of job passion was found to significantly enhance the positive impact of well-being on performance under conditions of high passion. In the Section 5, we will explore the implications of these findings for contemporary audit practices based on the latest research ([31]), offering evidence-based managerial recommendations to enhance the well-being and efficiency in auditing work.

## 2. Literature Review and Hypotheses Development

To ensure the timeliness and innovation of this study within the academic field, we have reviewed recently published literature, particularly studies released in recent years. These studies provide new insights, especially regarding the relationships between audit performance, job satisfaction, and the interplay of emotions and job performance. This not only enriches our theoretical foundation but also enhances the contemporary relevance of our research hypotheses.

### 2.1. Audit Performance and Job Well-Being

Audit performance is defined as an auditor’s performance or achievement of given audit tasks over a given period of time ([27]). Subjective well-being (SWB) is defined as the subjective perceptions of affect and life satisfaction ([15]). Although adjustments have improved the argumentation surrounding the construct of subjective job well-being, we acknowledge the lack of recent studies employing a similar approach. We review and integrate the contemporary literature discussing the nuances of subjective well-being in professional contexts, particularly in auditing and the related fields. This includes a review of the recent publications by prominent researchers such as ([13]), to ensure that the most current perspectives are incorporated.

In different societies, subjective well-being has both universal and unique predictors. Higher subjective well-being is associated with better job performance and creativity ([13]). The concept of subjective well-being at work should be elaborately described in terms of its direct impact on audit performance, including the aspects of job satisfaction, and positive and negative emotional responses. Moreover, recent research indicates a significant correlation between passion and job well-being ([32]), where the ongoing motivation provided by passion enhances individual career commitment and job satisfaction. Affect includes positive affect and negative affect. The experience of feeling happy, satisfied, or pleased is referred to as positive affect, and the experience of feeling upset, frustrated, or anxious is referred to as negative affect. Generally, people with more positive affect, less negative affect, and higher life satisfaction tend to have higher well-being ([14]). Based on the definition of SWB, this study defines job well-being as consisting of three components, including job satisfaction, positive affect, and negative affect. Most extant studies suggest that job well-being and job performance are in a positive linear relationship. This means that job performance increases or decreases in relation to job well-being ([22]; [23]; [26]; [36]; [47]).

A prior study employed affective events theory to investigate the effects of employee emotions on job performance ([2]), and found that the initial emotional reactions induced by job characteristics or requirements usually do not persist for a long time, and employees’ job performance varies due to fluctuating emotions. However, with positive characteristics and requirements continuing to exist and exert their influences in the workplace, employees’ emotions will develop into stable positive affect, which can, in turn, bring unexpected job performances. Auditors’ job is characterized by a high workload and responsibility. They are required to comply with standards on accounting and auditing, and complete assigned tasks within the specified time, so they are performing a highly professional job. Based on the view of Ashkanasy and Daus, this study argues that when auditors are influenced by positive events at work to show more positive emotions and fewer negative ones, their job well-being will grow. Because their emotions are still in an unstable state, their job well-being is usually not high, which would further affect their job performance. As a result, they will usually deliver a normal or even lower level of audit performance. However, with positive events at work continuing to emerge in the workplace, their job well-being will drastically increase. In the meantime, their pleasant emotions will induce a rapid growth of audit performance. To sum up, on the spectrum of job well-being from low to high, audit performance will slightly decrease first and then drastically increase later. As shown in Figure 1, the job well-being and audit performance are in a curvilinear relationship instead of a simple linear relationship. Therefore, the first hypothesis is proposed as follows:

**H1.** 
*Job well-being and audit performance are in a curvilinear relationship.*


### 2.2. The Effect of Passion on the Relationship Between Job Well-Being and Audit Performance

Passion is closely related to personality (e.g., extroversion) ([46]). People with passionate character will show passion when they immerse themselves in work ([7]; [45]). Passion is a momentum that can potentially drive actions. The positive momentum created by passion can help people achieve a good job performance ([3]; [17]). Due to high work stress, auditors in the Big Four CPA firms in Taiwan may not effectively improve their job performance without greater passion for work ([27]). We argue that for auditors with job well-being, greater passion for work can amplify the effect of job well-being, resulting in higher audit performance. In contrast, with low passion for work, the effect of well-being would be weakened. Therefore, auditors’ passion for work has a moderating effect on the relationship between job well-being and audit performance. More specifically, high passion for work can strengthen the effect of job well-being on audit performance, while low passion for work will dilute or eliminate the effect of job well-being on audit performance. Accordingly, the study proposes the following two hypotheses and illustrates the relationship of the three variables in Figure 2. The second hypothesis (H2) proposed in this study examines the moderating effect of auditors’ work passion on the relationship between job well-being and audit performance. Specifically, we hypothesize that if auditors possess high levels of work passion, the positive impact of job well-being on audit performance will be amplified. Conversely, if work passion is low, this positive impact will be diminished. The third hypothesis (H3) further refines this moderating effect, positing significant differences in the impact of job well-being on audit performance under conditions of high versus low passion. The distinction between these two hypotheses lies in that H2 establishes the moderating role of passion, while H3 delineates how this role manifests across different levels of passion.

**H2.** *There is a moderating effect of auditors’ passion on the relationship between job well-being and audit performance*.

**H3.** 
*High passion amplifies the effect of job well-being on audit performance; in contrast, low passion weakens the effect of job well-being on audit performance.*


## 3. Method

### 3.1. Participants

The sample population was expected to include all the employees in the auditing department of the Big Four CPA firms in Taiwan. To ensure that the participants had a sufficient understanding of their work environment and organizational atmosphere, we limited our subjects to auditors in a position of in-charge, deputy manager, and manager. The questionnaire survey method was adopted. The questionnaires were administered and collected with the assistance of the managers of the human resources in each CPA firm during the period of June 2022. A total of 550 copies were administered, and 185 responses were collected. Excluding 7 invalid responses, the final sample consisted of 178 responses. The valid response rate was 32.4%. Of the valid responses, 60% were males, 76% had three to five years of service duration, and 35% had a CPA license. All of the participants reported having a college or higher education degree. This study adhered strictly to the ethical standards in the collection and processing of data, ensuring the security and privacy of all the participants. Informed consent was obtained from all the participants prior to the study, and data were anonymized to prevent any breach of personal information.

### 3.2. Variables and Measures

The study included job well-being, audit performance, and passion as the main research variables and considered service length and gender as the control variables. To ensure the reliability and validity of each measurement scale, we obtained all the measures from prior studies or modified them based on our research design. The measurement scale of each variable is explained as follows.

#### 3.2.1. Job Well-Being

According to prior research, subjective well-being is defined as an individual’s emotional and cognitive evaluations of their life, encompassing life satisfaction and emotional experiences. The literature suggests that enhancing employees’ subjective well-being in the workplace is invaluable, as both subjective well-being and physiological well-being are crucial for both employees and organizations. Employee subjective well-being is associated with a broad range of work outcomes, including job performance ([41]), decisions to leave ([11]), organizational citizenship behaviors ([5]; [18]), and hostile acts towards colleagues and supervisors ([20]). In this study, “job well-being” is applied as a specific manifestation of subjective well-being within the workplace context, specifically referring to the emotional responses and job satisfaction of accountants at work. This distinction allows us to more precisely investigate the impact of the work environment on accountants’ emotions and performance. The term “subjective well-being” is consistently used to describe the broad concept of well-being, whereas “job well-being” is specifically employed when discussing aspects directly related to work.

Regarding the measurement tools used, the scales adapted from those developed for assessing subjective well-being have been employed to measure job well-being, which includes auditors’ job satisfaction, positive affect, and negative affect. This scale design enables us to accurately capture the specific emotional states within the workplace and their effects on job performance. Job satisfaction is rated with a scale adapted from the satisfaction with life scale developed by prior research ([12]), which has good reliability and validity ([39]; [40]), and has been used extensively ([16]). The adapted scale consists of 5 questions, with examples such as “Generally, my current work is very close to my ideal work” and “I have obtained what I have wanted from my job”. All the questions were designed to be rated on a 7-point Likert scale (from 1-Very disagree to 7-Very agree). In addition, positive affect and negative affect are measured using a scale developed by prior research ([6]). In this scale, positive affect and negative affect were measured based on the emotional response of the participants over the past few weeks. Each uses five questions on a 7-point Likert scale (from 1-Never to 7-Always). The scores for job satisfaction, positive affect, and negative affect (reverse scoring) are summed. The higher the total score, the higher the job well-being. The Cronbach’s alpha for the three scales is 0.78, 0.80, and 0.82, respectively, and for the total scale is 0.80.

#### 3.2.2. Audit Performance

Audit performance is the degree of achievement of audit tasks. To measure audit performance, the study developed a scale based on the prior research ([28]), which has been used extensively ([34]). The scale consists of 8 questions covering planning, investigation, coordination, review, supervision, human resource utilization, external relations, and representation for departments, for instance, “My performance on planning audit goals and task progress” and “My performance on information processing and investigation”. All the questions were to be rated on a 7-point Likert scale (from 1-Very poor to 7-Very good). A higher total score indicates a higher audit performance. The Cronbach’s alpha for this scale is 0.86.

#### 3.2.3. Passion

Passion was measured by using the harmonious passion scale developed by a prior study ([45]), which has been widely used by prior passion studies ([43]) and modified according to the research context. The final scale consisted of 7 questions, including “My work allows me to live a variety of experiences” and “The new things that I discover with my work allow me to appreciate it even more”. All the questions are also rated on a 7-point Likert scale. A higher total score indicates higher passion. The Cronbach’s alpha for this scale is 0.82.

#### 3.2.4. Control Variables

This study mainly explored the effects of emotional reactions in regard to job well-being and passion on auditors’ audit performance. Because an auditor’s emotional reaction may be affected by their job experience ([10]) and gender ([25]), this study thus considers service length and gender as control variables.

### 3.3. Measurement Model and CFA

The study used LISREL as an analytic tool to examine the measurement model and conduct the confirmatory factor analysis (CFA). The fit indices of the measurement model are as shown in Table 1. The full model refers to the measurement model including all the items of job well-being, audit performance, and passion. As shown in Table 1, the goodness of fit of the model is not good (χ2/d.f. = 1503.19/403 = 3.73, GFI = 0.72, AGFI = 0.68, NFI = 0.74, NNFI = 0.75, CFI = 0.81, RMSEA = 0.24). The estimation with the maximum likelihood method showed that the factor loading of some items did not reach a significant level (i.e., item 5 for job satisfaction, item 4 for positive affect, item 5 for negative affect, items 6 and 8 for audit performance, and item 6 for passion). The modified model is the model excluding these insignificant items. Judging by the fit indices, the goodness of fit of the modified model is good (χ^2^/d.f. = 464.35/251 = 1.85, GFI = 0.91, AGFI = 0.91, NFI = 0.97, NNFI = 0.97, CFI = 0.98, RMSEA = 0.05). Therefore, the study used the modified model for CFA.

As shown in Table 2, the CFA results indicate that all the items have a factor loading that is greater than 0.5 and significance in the t-test, suggesting the convergent validity of all the variables ([1]). All the average variance extracted (AVE) values were above 0.5, and all the composite reliability values were greater than 0.7, meaning that all the variables have an acceptable level of internal consistency. As shown in Table 3, the square root of AVE for each variable is greater than the correlation of that variable with other variables, suggesting discriminant validity between the variables ([4]). Through the above analyses, the study confirmed the unidimensionality of all the measurement variables.

### 3.4. Research Model

The proposed hypotheses were tested using two regression models as follows:

Model 1:Audit performance = *β*0 + *β*1 job well-being + *β*2 job well-being^2^ + *β*3 passion + *β*4 service length + *β*5 gender + *e*

Model 2:Audit performance = *β*0 + *β*1 job well-being + *β*2 passion + *β*3 job well-being × passion + *β*4 service length + *β*5 gender + *e*

In Model 1, squared job well-being (job well-being^2^) is used to evaluate the nonlinear relationship between job well-being and audit performance. If the *β*2 coefficient of job well-being^2^ is significant, the nonmonotonic effect of job well-being on audit performance is significant. This means that the relationship between job well-being and audit performance is curvilinear, and H1 is supported.

Model 2 is designed to test the moderating effect of passion, that is, whether job well-being and passion have an interaction effect on audit performance. In this model, the *β*3 coefficient of the product term of job well-being and passion is used to test the interaction effect. A positive and significant *β*3 proves the moderating effect of passion on the job well-being–audit performance relationship, and H2 is supported. In order to further analyze the effect of an auditor’s passion, this study divided the passion into high and low groups, and performed regression analysis, respectively, to test the relationship between job well-being and audit performance to verify the support for H3.

## 4. Results and Analyses

### 4.1. Descriptive Statistics

The correlation coefficients between the variables and descriptive statistics of each variable are, respectively, shown in Table 3 and Table 4. As shown in Table 3, there is a quite significant correlation between audit performance and job well-being (r = 0.44, *p* < 0.01), but the correlation between passion and job well-being is not very significant (r = 0.20, *p* < 0.05).

### 4.2. The Effect of Job Well-Being on Audit Performance

To gain an insight into how job well-being affects audit performance, we conducted a regression analysis of the effects of job well-being, job well-being^2^, passion, and control variables on audit performance. As shown in Table 5, the coefficient of job well-being^2^ is significant (*β*_2_ = 0.50, *p* < 0.01). This suggests that the effect of job well-being on audit performance is nonmonotonic. This finding conforms to our assumption that job well-being and audit performance are in a curvilinear relationship. Thus, H1 is supported.

### 4.3. The Moderating Effect of Passion

To examine whether the effect of job well-being on audit performance is moderated by passion, we conducted a regression analysis of the effects of job well-being, passion, the product term, and control variables on audit performance. As shown in Table 6, the interaction term of job well-being and passion is significant (*β*_3_ = 0.62, *p* < 0.01), meaning that passion has a moderating effect on the relationship between job well-being and audit performance. H2 is thus supported.

To further clarify the interaction effect, the total samples were divided by the median value of passion ([33]) into a high passion group and low passion group. The findings from the high passion group and the low passion group are, respectively, presented in Panel A and Panel B of Table 7. As shown in Panel A, in the high passion context, the positive correlation between job well-being and audit performance is very significant (*β*_1_ = 0.56, *p* < 0.01). In Panel B, the low passion condition, such correlation becomes insignificant (*β*_1_ = 0.16, *p* > 0.1). These findings are in line with the relationship illustrated in Figure 2. In other words, for auditors with high passion for work, higher job well-being can lead to higher audit performance, but for auditors with low passion for work, the job well-being cannot contribute to an improvement in audit performance. Thus, H3 is supported.

## 5. Discussion

The study conducted an in-depth analysis of the results based on the contemporary research, critically examining the implications of our findings within the broader fields of subjective well-being and job performance. This discussion not only integrates the main findings from the field but also updates and expands the previously proposed theoretical framework to include a broader range of the literature and current research developments.

### 5.1. Overview of the Key Findings

Our research confirms a curvilinear relationship between job well-being and performance, moderated by the variable of passion. This finding aligns with recent theories suggesting that the dynamics of well-being are more complex than traditionally understood, involving multiple interactive factors that can vary significantly across different professional environments ([13]). Moreover, this finding highlights the potential application of the research in actual work settings, especially in the formulation of human resource management and organizational behavior strategies.

### 5.2. Theoretical Implications

Confirming the curvilinear relationship contributes to the advancement of psychological research, particularly in understanding the nonlinear impacts of emotional states on productivity. In recent years, the management literature has begun to discuss the individual and organizational outcomes following happiness, rather than the pursuit of happiness and its ultimate achievement in business life ([24]). Following the fact that overdosing on any substance can be harmful, it has become the subject of more research, with results that are contradictory, indicating that the happiness that evokes positive emotions does not always lead to the same outcomes.

This challenges the traditional models and supports a more nuanced view presented in recent research. These theoretical implications are crucial for designing programs to enhance well-being and workplace strategies, advising managers and policymakers to consider the multidimensional nature and impact of well-being. These discussions extend our understanding of the relationship between job well-being and performance and promote the research on how these insights can be effectively applied across various professional backgrounds. Through these in-depth discussions, we hope to provide a more comprehensive perspective for assessing and addressing the issues of well-being and passion in the workplace, ultimately enhancing the overall well-being of employees and organizational performance.

## 6. Conclusions

Initially, this study substantiated the hypothesis that a curvilinear relationship exists between auditors’ job well-being and audit performance, confirming that job well-being significantly influences the audit outcomes, albeit in a non-linear manner. Specifically, our research findings reveal that higher levels of job well-being are associated with rapid improvements in audit performance, highlighting the crucial role of well-being in the auditing profession ([23]; [47]). Moreover, our analysis supports the hypothesis of a synergistic interaction between job well-being and passion, significantly enhancing audit performance when both are elevated. Conversely, low passion appears to negate the positive effects of high job well-being on performance, aligning with previous research that posits a conditional relationship dependent on the levels of passion. This nuanced comprehension affirms that the optimal impact of job well-being on performance predominantly emerging under conditions of high passion, thus offering a comprehensive view of the dynamics between emotional well-being and professional efficacy within audit organizations.

Our results offer a number of implications as follows: First, when job well-being is low, audit performance will not increase and may even decline. However, with the increase in job well-being, there will be a rapid growth in audit performance. This suggests that for audit organizations that stress professional services, strengthening job well-being can lead to a performance increase that is more than proportional to the increase in job well-being and may even exceed expectations. Additionally, auditors are required to show high efficiency and effectiveness at work. In addition to job well-being, their passion for work is also critical to improving the audit performance. On the other hand, even if they have high passion for work, without job well-being, the effect of their passion will be inhibited, and their audit performance may not necessarily improve. This finding implies that while creating a job well-being environment, audit organizations should also manage to increase employees’ passion for work, so that the two complement each other so as to effectively improve the overall audit performance. Other professional organizations, such as hospitals and law firms, are also professionally oriented. Most employees in these organizations have long working hours, and their services are usually closely related to the lives and properties of citizens. Their importance is definitely of no doubt. Therefore, for higher organizational performance, these professional organizations are advised to improve their employees’ passion and job well-being at the same time.

The contributions of this study and suggestions for future research are summarized as follows: First, most previous research indicates that job well-being and organizational performance are in a linear relationship. However, our evidence shows that the relationship is actually curvilinear. When auditors have low job well-being, their audit performance will not be very high. With the increase in job well-being, there will be a quick rise in audit performance. We not only challenge the finding of previous research but also offer empirical evidence on the curvilinear relationship between job well-being and audit performance. Second, previous research has only explored the effect of employees’ passion for work. In this paper, we further divide passion into high passion and low passion to more comprehensively capture the effect of passion on audit performance. Third, more and more businesses are committed to creating a positive work environment that can increase employees’ job well-being. In this paper, we find that high job well-being cannot lead to higher audit performance without high passion for work. Finally, employees in professional organizations are different in many aspects from employees in other types of organizations. In terms of the roles in the organization and contributions, they should be discussed separately from general employees. In this paper, we focus on professional audit organizations. Our findings fill a gap in the literature and also provide a reference for the behavioral research of professional employees. As for suggestions for future research, the behavioral model of auditors can be further discussed. For instance, organizational identity can be included in the model to obtain more in-depth and comprehensive results. In addition, the subjects in this study are auditors in the Big Four CPA firms in Taiwan. Future research can also include employees in non-Big Four CPA firms or compare the differences between them. Finally, audit performance was determined through self-evaluation by the participants. Future research can adopt a multi-assessment approach by both auditors and their direct supervisors to determine the audit performance.

The study has its limitations. The research data came from the auditors of CPA firms. Their workload is affected by the firm’s light season or busy season. Therefore, the work emotion and psychological cognition are different in different work periods. The research period was carried out during the light season, so the research results may differ from those from the busy season. Moreover, data were collected only one time from the same source, so there may be a common method variance (CMV) problem. However, because the research variables have convergent validity, discriminant validity, and unidimensionality, the CMV should not seriously affect the results of the study. Furthermore, this study was conducted within the auditing environment in Taiwan, providing us the opportunity to explore how cultural background shapes job well-being and its impact on audit performance. The work culture in Taiwan emphasizes teamwork and harmony, which may influence auditors’ perceptions of job satisfaction and professional passion. Further research could compare how auditors from different cultural backgrounds experience work stress and satisfaction, thereby offering a more comprehensive perspective to understand the relationship between job well-being and audit performance.

## Figures and Tables

**Figure 1 behavsci-15-00042-f001:**
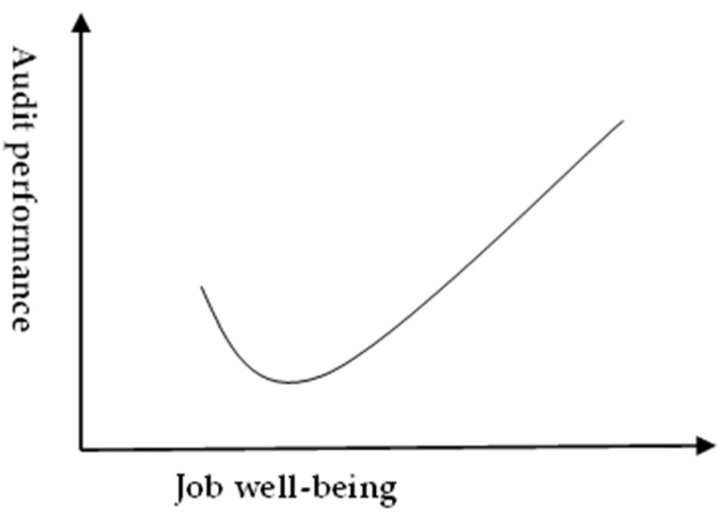
The relationship between job well-being and audit performance.

**Figure 2 behavsci-15-00042-f002:**
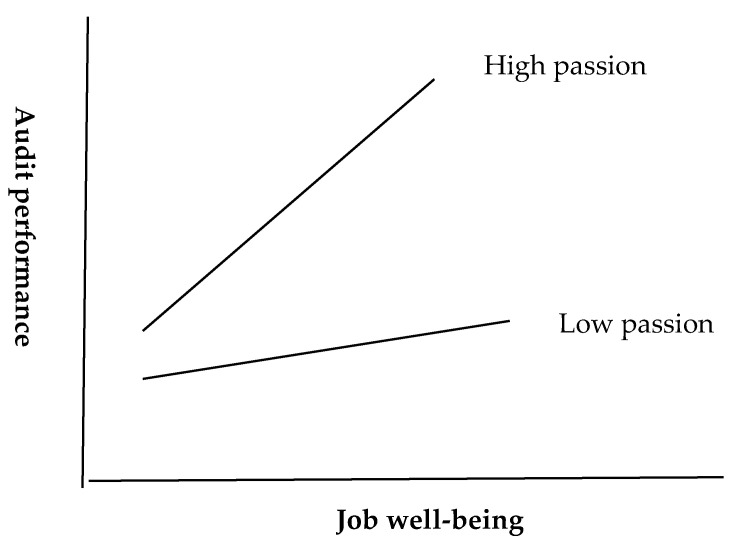
The relationship among passion, job well-being, and audit performance.

**Table 1 behavsci-15-00042-t001:** Goodness of fit of the measurement model.

Model	χ^2^	d.f.	GFI	AGFI	NFI	NNFI	CFI	RMSEA
Full model	1503.19	403	0.72	0.68	0.74	0.75	0.81	0.24
Modified model	464.35	251	0.91	0.91	0.97	0.97	0.98	0.05

**Table 2 behavsci-15-00042-t002:** Measurement validity and reliability.

Variable	Item	Factor Loading	Composite Reliability	AVE
Job well-being			0.81	0.55
	jwb1	0.73		
	jwb2	0.61		
	jwb3	0.66		
	jwb4	0.64		
	jwb6	0.75		
	jwb7	0.83		
	jwb8	0.77		
	jwb10	0.68		
	jwb11	0.57		
	jwb12	0.65		
	jwb13	0.78		
	jwb14	0.70		
Audit performance			0.84	0.53
	ap1	0.65		
	ap 2	0.72		
	ap 3	0.66		
	ap 4	0.67		
	ap 5	0.82		
	ap 7	0.75		
Passion			0.83	0.52
	pa1	0.66		
	pa2	0.77		
	pa3	0.70		
	pa4	0.81		
	pa5	0.78		
	pa7	0.63		
Notes: χ^2^ = 464.35, d.f. = 251, GFI = 0.91, AGFI = 0.91, NFI = 0.97, NNFI = 0.97, CFI = 0.98, RMSEA = 0.05.				

**Table 3 behavsci-15-00042-t003:** Pearson correlation coefficients.

Variable	Job Well-Being	Audit Performance	Passion
Job well-being	(0.74) ^a^		
Audit performance	0.44 **	(0.73)	
Passion	0.20 *	0.11	(0.72)

^a^ Values within parentheses are the square root of AVE. ** *p* < 0.01; * *p* < 0.05.

**Table 4 behavsci-15-00042-t004:** Descriptive statistics.

Variable	Theoretical Range	Min	Max	Mean	S. D.
Job well-being	12–84	17	83	51.72	10.15
Audit performance	6–42	9	42	26.36	5.06
Passion	6–42	8	41	25.18	4.52
Service length	-	2	8	4.82	1.24
Gender	1–2	1	2	1.39	0.47

**Table 5 behavsci-15-00042-t005:** Regression analysis of the effect of job well-being on audit performance.

Variable	Coefficient	Estimate	Standard Error	t Value	*p* Value
Constant	*β* _0_	21.25	3.30	6.37	0.00 **
Job well-being	*β* _1_	0.17	0.07	2.00	0.03 *
Job well-being^2^	*β* _2_	0.50	0.11	3.81	0.00 **
Passion	*β* _3_	0.13	0.06	1.51	0.12
Service length	*β* _4_	0.02	0.04	0.42	0.50
Gender	*β* _5_	0.01	0.01	0.51	0.48
R^2^ = 0.46, F = 11.37 **					

Dependent variable: audit performance; ** *p* < 0.01, * *p* < 0.05.

**Table 6 behavsci-15-00042-t006:** Regression analysis of the effect of job well-being and passion on audit performance.

Variable	Coefficient	Estimate	Standard Error	t Value	*p* Value
Constant	*β* _0_	23.77	5.41	4.40	0.00 **
Job well-being	*β* _1_	0.38	0.14	2.53	0.01 **
Passion	*β* _2_	0.12	0.06	1.86	0.08
Passion × Job well-being	*β* _3_	0.62	0.22	2.70	0.01 **
Service length	*β* _4_	0.40	0.24	1.54	0.14
Gender	*β* _5_	0.01	0.02	0.42	0.75
R^2^ = 0.45, F = 13.68 **					

Dependent variable: audit performance; ** *p* < 0.01.

**Table 7 behavsci-15-00042-t007:** Decomposition of the interaction effect of passion and job well-being.

Variable	Coefficient	Estimate	Standard Error	t Value	*p* Value
Panel A: High passion (n = 89)					
Constant	*β* _0_	11.49	4.16	2.86	0.01 **
Job well-being	*β* _1_	0.56	0.11	5.43	0.00 **
Service length	*β* _4_	0.35	0.15	2.25	0.04 *
Gender	*β* _5_	0.13	0.10	1.31	0.18
R^2^ = 0.33, F = 6.34 **					
Panel B: Low passion (n = 89)					
Constant	*β* _0_	4.37	2.03	2.13	0.03 *
Job well-being	*β* _1_	0.16	0.13	1.16	0.35
Service length	*β* _4_	0.14	0.08	1.31	0.18
Gender	*β* _5_	0.13	0.07	1.24	0.21
R^2^ = 0.12, F = 2.17 *					

Dependent variable: audit performance; ** *p* < 0.01, * *p* < 0.05.

## Data Availability

Research data are available from the corresponding author with the consent of all authors.

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
