# Peer review of "How Does Job Well-Being Optimize Audit Performance? The Moderating Effect of Passion"

_behavsci, 2025, doi:10.3390/bs15010042_

Round 1

Reviewer 1 Report

Comments and Suggestions for Authors

Introduction:

The authors should provide a more detailed explanation of the job characteristics and requirements that can be examined through the lens of affective events theory. This clarification may enhance the understanding of the relationship between auditors’ job characteristics and the study’s proposition regarding a curvilinear relationship between job well-being and audit performance. I recommend that the authors delve deeper into the clarification of the core concepts. Furthermore, while the study was conducted in Taiwan and the majority of the company examples provided are also from this context, the authors fail to contextualize the cultural factors underlying the study. Is there a variation in the perception of job well-being among employees from different cultural backgrounds?

Literature Review:

This section raises a significant concern regarding the definition of job well-being. I did not find the construct to be adequately supported within the existing literature. The authors have merged perceptions of affect and life satisfaction; however, there is no discussion of other theoretical frameworks that explore similar concepts, such as Job-related Affective Well-Being Scale (JAWS) or the job well-being framework proposed by Chen and Li (2017)*. If the intention is to develop a new scale for the job well-being construct based on existing variables, it is imperative to adopt different methodological procedures.

Another aspect that requires further clarification is the distinction between Hypothesis 2 (H2) and Hypothesis 3 (H3). Given the results of H2, could the authors draw any conclusions regarding H3?

Lastly, a major concern I have is that the references cited are outdated. The most recent paper referenced is from 2021, which falls short of current academic standards. I strongly recommend that the authors update the literature review.

*Chen, H., & Li, G. (2017). A review of factors influencing nurses' intention to leave and interventions. Journal of Nursing Science, 32(12), 106–108.

Method:

The measurement of job well-being remains a concern. Are there any prior studies that have employed these three components to assess job well-being? The strategy of summing the scores should be substantiated with supporting literature. This is something that the authors realy need to work on.

I recommend to include a table summarizing the results of the measurement model in terms of convergent and discriminant validity, as well as reliability.

Results and Discussion:

First, the issues associated with the job well-being construct must be adequately addressed. Furthermore, the discussion section needs to be significantly enhanced, as it currently lacks substance (it appears to be integrated with the conclusion section). The authors should discuss the results in relation to relevant prior studies and findings, as well as within the context of the chosen theoretical framework. Additionally, caution should be taken when making generalizations to other professional fields, such as law or medicine, as these were not considered in the study. Once again, I believe that cultural factors should be better incorporated into the discussion.

Formatting:

  • Add sources to figures and tables.
  • Correct inconsistencies in reference formatting; it is preferable to include DOIs in the references.
  • Update reference list.
  •  
Comments on the Quality of English Language
  •  
  • Rectify minor issues in English language usage.

Reviewer 2 Report

Comments and Suggestions for Authors

What confuses me in this paper is that the authors use the term subjective well-being (SWB) while in fact defining job well-being (80-82; 170-184). It would still be more correct in this paper to use the term/construct Subjective Well-Being instead of Well-Being. “Subjective well-being (SWB) is the personal perception and experience of positive and negative emotional responses and global and (domain) specific cognitive evaluations of satisfaction with life.” Proctor, C. (2014). Subjective Well-Being (SWB). In: Michalos, A.C. (eds) Encyclopedia of Quality of Life and Well-Being Research. Springer, Dordrecht. https://doi.org/10.1007/978-94-007-0753-5_2905

 SWB has been defined as “a person’s cognitive and affective evaluations of his or her life”. Diener, E., Lucas, R. E., & Oishi, S. (2002). Subjective well-being: The science of happiness and life satisfaction. In C. R. Snyder & S. J. Lopez (Eds.), Handbook of positive psychology (pp. 463–73). Oxford University Press. p. 63).”

Starting from the author Diener, who is quoted in the paper (80), it is evident that he analysed (construct) subjective well-being:

Diener, E. (1984). Subjective well-being. Psychological Bulletin, 95, 542–575.

Diener, E. (1994). Assessing subjective well-being: Progress and opportunities. Social Indicators Research, 31, 103–157.

Diener, E. (1996). Traits can be powerful, but are not enough: Lessons from subjective well-being. Journal of Research in Personality, 30, 389–399.

Diener, E., Diener, M., & Diener, C. (1995). Factors predicting the subjective well-being of nations. Journal of Personality and Social Psychology, 69, 851–864.

Diener, E., & Suh, E. M. (1998). Subjective well-being and age: An international analysis. In K. W. Schaie & M. P. Lawton (Eds.), Annual review of gerontology and geriatrics (Focus on emotion and adult development, Vol. 17, pp. 304–324). New York: Springer.

Diener, E., Lucas, R. E., & Oishi, S. (2002). Subjective well-being: The science of happiness and life satisfaction. In C. R. Snyder & S. J. Lopez (Eds.), Handbook of positive psychology (pp. 63–73). New York: Oxford University Press.

Diener, E., & Suh, E. M. (1999). National differences in subjective well-being. In D. Kahneman, E. Diener, & N. Schwarz (Eds.), Well-being: The foundations of hedonic psychology (pp. 434–450). New York: Sage.

Diener, E., Suh, E. M., Lucas, R. E., & Smith, H. L. (1999). Subjective well-being three decades of progress. Psychological Bulletin, 125, 276–302.

Diener, E., Suh, E. M., & Oishi, S. (1997). Recent findings on subjective well-being. Indian Journal of Clinical Psychology, 24, 25–41.

Diener, E., Suh, E. M., Smith, H., & Shao, L. (1995). National differences in reported subjective well-being: Why do they occur? Social Indicators Research Special Issue: Global Report on Student Well-Being, 34, 7–32.

Eid, M., & Diener, E. (2004). Global judgments of subjective well-being: Situational variability and long-term stability. Social Indicators Research, 65, 245–277.

Emmons, R. A., & Diener, E. (1985). Personality correlates of subjective well-being. Personality and Social Psychology Bulletin, 11, 89–97.

We can assume that subjective well-being (SWB) and Quality of Life (QOL) converge. Nowadays, organizations like Deloitte, for example, adopt a Wellbeing Strategy that promotes a holistic approach to employee well-being (See: https://www.deloitte.com/cy/en/careers/deloitte-life/benefits/well-being.html). Employee well-being is determined by the balance between work and private life, good physical health, reduced stress and enriched life. Furthermore, the 2022 "The U.S. Surgeon General's Framework for Workplace Mental Health & Well-Being" describes “Five Essentials for Workplace Mental Health & Well-Being” (See: chrome-extension://efaidnbmnnnibpcajpcglclefindmkaj/https://www.hhs.gov/sites/default/files/workplace-mental-health-well-being.pdf):

1.      Protection from Harm (Safety, Security)

2.      Connection & Community (Social Support, Belonging)

3.      Work-Life Harmony (Autonomy, Flexibility)

4.      Mattering at Work (Dignity, Meaning)

5.      Opportunity for Growth (Learning, Accomplishment)

 We have a number of such business cases, but independently of that, if we analyse the results of previous scientific research, I cannot explain why the authors assessed job satisfaction with a scale adapted from the life satisfaction scale developed by Diener, Emmons, Larsen and Griffin (1985), when there are scales such as:

Watanabe K, Imamura K, Inoue A, Otsuka Y, Shimazu A, Eguchi H, et al. Measuring eudemonic well-being at work: a validation study for the 24-item the university of Tokyo occupational mental health (TOMH) well-being scale among Japanese workers. Ind Health. (2020) 58:107–31. 10.2486/indhealth.2019-0074

Zheng, X., Zhu, W., Zhao, H., Zhang, C. (2015). Employee well-being in organizations: Theoretical model, scale development, and cross-cultural validation. Journal of Organizational Behavior, 36, 621-644. “The Employee Well-Being Scale is an 18-item scale comprised of three facets of well-being: life well-being (LWB), work well-being (WWB), and psychological well-being (PWB) and contains 6 items for each domain.” https://www.hsph.harvard.edu/health-happiness/employee-well-being-scale/

(Demo, Gisela & Paschoal, Tatiana. (2016). Well-Being at Work Scale: Exploratory and Confirmatory Validation in the USA. Paidéia (Ribeirão Preto). 26. 35-43. 10.1590/1982-43272663201605.)

Laila Leite Carneiro, Antônio Virgílio Bittencourt Bastos. (2023) Escala de bienestar en el trabajo (EBET): compilación de evidencia empírica de

diferentes estrategias de interpretación. (Well-being at work scale (WBWS): Gathering empirical evidence from different interpretation strategies), Quaderns de Psicologia, Vol. 25, Nro. 1

In the section Methods (150-162) the research period is missing.

Reviewer 3 Report

Comments and Suggestions for Authors

How does Job Well-Being Optimize Audit Performance? The Moderating Effect of Passion

The topic is relevant to people management and organizational results. Opportunities for improvement are presented to reinforce the quality, support and discussion of the article presented:

Abstract

• Should include information about the research approach adopted and the number of participants. It will also be important to briefly clarify the practical implications of the results obtained.

Introduction

• The case presented as an auditing company with a well-being policy has a bibliographic reference from 2004. It is important to present a more current characterization of the context.

• It will be important to strengthen, with bibliographical support, the premise that the relationship between well-being and performance is not linear, but curvilinear.

• The framing of the passion construct is supported by literature over 8 years old, and most of it over 20 years old. It is important to show a more current review of the literature.

Theoretical Structure

• More space is given to the definition of subjective well-being in life than to well-being at work, which is the construct under analysis. Reinforce this concept.

• There is still a lack of solid contextualization for the initial premise that the auditor, at an early stage, has reduced performance and well-being

• It is necessary to reinforce the theoretical framework associated with passion with more recent references

• Reference appears to studies that link passion and performance, but nothing is mentioned regarding the relationship between passion and well-being at work. Complete, please.

Method

• Important to add a section on ethical issues observed in data collection, processing and storage

Conclusions

• It is important to start by systematizing the results obtained in relation to the hypotheses stated

• The first paragraph is the only one that calls on the literature to debate the results, but it does so in a generic way and without addressing two issues identified in the hypotheses: the curvilinear relationship and the moderate effect of passion

• The discussion must have an independent chapter with recent literature. The current references are more than 20 years old and are redundant to those cited in the theoretical framework.

• Implications for practice must be supported by bibliographic references. The authors limit themselves to their own understanding, making their proposals fragile.

I hope these suggestions help the authors in their work to improve the current version of the article.

Round 2

Reviewer 1 Report

Comments and Suggestions for Authors

The authors have made significant improvements in addressing the majority of the issues raised in the initial review, and I commend them for their efforts. However, I believe there are still some aspects that require further clarification.

Despite the modifications introduced, the first point in the literature review regarding the theoretical support for the concept of subjective job well-being has not yet been adequately addressed. While the authors improved the argumentation surrounding the construct, they did not provide sufficient theoretical support (references) or examples of previous studies that used a similar approach to assess subjective well-being. Furthermore, the references to the work of Diener, E., are outdated, as the author has conducted more recent research on these topics.

Regarding the results for convergent and discriminant validity, the findings were summarized, but I would prefer to see the detailed results for each variable to allow for comparisons with similar studies.

Author Response

The authors are grateful to the reviewer for once again providing very constructive comments that helped improve this article. The authors respond to the reviewer's comments point-to-point and revise the article. The response content clearly indicates the revised parts in the article, both of which are displayed in green fonts. Please refer to the attachment for details, thank you again.

Reviewer 2 Report

Comments and Suggestions for Authors

The authors explained the term and stated the choice in lines 205-217, but still do not explain why they chose that particular approach (perhaps it fits better in a theoretical model or framework?). The authors still did not indicate that there are alternative questionnaires and critically reviewed the advantages and disadvantages to justify their approach.

If the validated questionnaire is adapted, it should be validated again. Any adaptation can affect the structure, clarity, reliability and validity of the questionnaire, so it is necessary to carry out additional checks to ensure that the adapted version retains the same qualities as the original (see: 218-220).

In lines 220-230, the authors give an example of two questions (without listing all the questions and only report Cronbach's alpha (internal consistency - reliability). Later, the authors conduct a CFA (provides construct validity; and this is very limited reporting; in lines 259-260, they only repeat; it would be good to specify the factor loadings). Is there a lack of criterion validity (does it correlate with related questionnaire items that measure the same or a similar concept)?

Author Response

(The authors gave the same response as above.)

Reviewer 3 Report

Comments and Suggestions for Authors

I appreciate all the changes made. There remains a need to reinforce the discussion of results as indicated in the previous review report:

Conclusions

• It is important to start by systematizing the results obtained in relation to the hypotheses stated

• The first paragraph is the only one that calls on the literature to debate the results, but it does so in a generic way and without addressing two issues identified in the hypotheses: the curvilinear relationship and the moderate effect of passion

• The discussion must have an independent chapter with recent literature. The current references are more than 20 years old and are redundant to those cited in the theoretical framework.

• Implications for practice must be supported by bibliographic references. The authors limit themselves to their own understanding, making their proposals fragile.

Author Response

(The authors gave the same response as above.)

Round 3

Reviewer 2 Report

Comments and Suggestions for Authors

I want to thank the authors for their answers and adopted suggestions.